

# Research on emotion classification technology of movie reviews based on topic attention mechanism and dual channel long short term memory

Yufei Wang

School of Film and Television Media, Wuchang University of Technology, Wuhan, Hubei, China

## ABSTRACT

In recent years, with the popularity of the Internet, more and more people like to comment on movies they have watched on the film platform after watching them. These reviews hide the reviewers' feedback on films. Mining the emotional orientation information in these reviews can provide consumers with shopping references and help businesses optimize film works and improve business strategies. Therefore, the emotional classification of film reviews has high research value because few emotion dictionaries and analysis tools are available for reference and use in film reviews. The accuracy of emotion classification still needs to be improved. This study introduces the attention mechanism and dual channel long short term memory (DC-LSTM) while building the emotion dictionary in the field of Chinese film review. It classifies Chinese film reviews in terms of topic-based fine-grained emotion. First, the emotion vector is constructed using the constructed movie review emotion lexicon. The semantic vector obtained by the Word2vector tool is input to LSTM to encode the comment text. Then, the topic attention module is used to decode. Finally, the final emotion classification result is obtained through the softmax function of the entire link layer and the output layer. The thematic attention modules constructed in this study are independent of each other for attention parameter adjustment and learning. One attention module corresponds to one film theme. In this study, eight themes, including "plot," "special effects," "original work," "music," "thought," "theme," "acting skills," and "joke," were extracted, and each theme was classified into three types of emotions: "positive," "neutral," and "negative." The experimental results on the crawled Chinese film review dataset show that the proposed algorithm is superior to some existing algorithms and models in accuracy, precision, recall and F1 measure. The DCLSTM based on the thematic attention mechanism (DCLSTM-TAM) model constructed in this study introduces the emotion vector into the network and adds the theme attention mechanism. It can not only classify the emotion for different topics of a film review but also effectively deal with film reviews with fuzzy emotional tendencies. It realizes the fine-grained emotion classification of film topics and improves the accuracy of emotion classification of film reviews. The emotion classification method and model proposed in this study have good transferability, and the change of training corpus is also applicable to other short text fields.

Corresponding author
Yufei Wang, 18167821632@163.com

## INTRODUCTION

China's film market and related industries have developed rapidly in recent years. More and more users choose to communicate their opinions and opinions through the network platform before and after watching the film, which has produced many informative texts. An online film review is a key factor affecting a film's box office. It is a comprehensive review of a film's plot, special effects, actors, sound effects, etc., by audiences or professional film reviewers, which contains rich emotional information. The emotional orientation classification of film reviews has rich commercial value, which has become an essential aspect of text emotion analysis and research and has also attracted the attention of many scholars.

The emotional analysis of an unstructured text, such as user evaluation and comments, can be divided into three categories according to the technical characteristics: rule-based emotional thesaurus matching analysis method, statistical machine learning-based emotional classification method, and machine learning method classification combined with dictionary filtering emotional analysis algorithm. The main idea of the emotion analysis research method based on an emotion dictionary is to calculate the emotional tendency of comments by matching a specific emotion dictionary after word segmentation. This method is widely used in the early stage of emotion analysis research and follows a basic rule: emotion analysis mostly depends on the merits of the emotion dictionary. The richer the emotion words contained in the emotion dictionary, the more accurate the result of emotion analysis. Because the process of this method is simpler and easier to use than that based on machine learning, many dictionary-based emotion analysis studies have emerged. *Turney & Littman (2003)* was the first to classify the emotions of reviews in the film field and proposed a method based on PMI-IR to calculate the emotional tendency of reviews. *Rodzin et al. (2022)* calculated the distance between the subject words and emotional words in the text to interfere with the emotional tendency of the comment and concluded that when the distance between the subject words and emotional words increases, the emotional tendency of the comment will weaken. *Li & Ji (2015)* proposed a new emotion analysis algorithm by integrating lexical analysis and emotion level dictionary and classified the objective and subjective datasets in the data set. *Pandarachalil, Sendhilkumar & Mahalakshmi (2015)* innovatively used MapReduce technology to construct emotional dictionaries, applied cloud computing technology to construct emotional lexicon, and improved emotional dictionaries' construction efficiency.

In addition, the method based on the emotion dictionary can not comprehensively analyze all the views in the comments, and it is only applicable to analyzing the emotional tendency in the comments. The classification models in machine learning, such as SVM and logical regression, can propose features according to specific algorithms to achieve better emotional classification effects, so they are widely used. *Calvo & D'Mello (2010)* studied the

construction of tagged data sets and the knowledge-based and corpus-based text emotion recognition methods related to automatic text emotion analysis. *Kaur & Gupta (2013)* used a hierarchical approach to recognize and classify text emotions by considering the hierarchical structure of neutral, polar, and emotions. *Xu et al. (2017)* proposed a multi-tag method to detect emotional factors, which can detect emotional words and sentences and capture cross-sentence information to detect emotional factors. *Jang et al. (2012)* combined emotional words, hashtags and emotions to achieve multi-category emotion classification by distance supervision. *Wang & Ester (2014)* proposed two emotional theme models to classify social emotions and build an emotional dictionary. *Campos et al. (0000)* used Unigram, Bigram and Trigram as emotion classification features and used a naive Bayesian algorithm to make emotional judgments on subjective sentences. *Wang, Zhao & Fu (2011)* applied the fuzzy set theory to the emotion classification task in the Chinese context and constructed a Chinese emotion classification framework based on the fuzzy set theory.

A film text review is the audience's most intuitive feeling and evaluation after watching the film. It is essential data to measure the value of a film, and its in-depth study has important reference and application value. However, the above research is still not enough to make valuable decisions, and there are still several shortcomings in practical applications: (1) the overall emotional analysis has failed to meet people's needs, especially for enterprises and consumers, they began to pursue more detailed and accurate emotional analysis. Enterprises hope to obtain the evaluation on specific features or attributes of their products or services from online comments to make targeted improvements; due to individual differences, consumers pay different attention to the same product or service, hoping to obtain personalized product information from massive comments, to make decisions quickly. (2) Although a sentence may have an overall positive or negative emotional tendency, the elements in the sentence may express opposite views. A positive evaluation of a product does not mean that the reviewer likes all the features or attributes of the product. Similarly, negative comments do not mean that the reviewer does not like everything. For example, the sentence "The plot of the film is old-fashioned, but the acting skills of the actors are good, and the overall performance is also good" evaluates two characteristics of the film (entity), namely, the plot and the acting skills of the actors. The emotional attitude towards the film plot is negative, but the emotional attitude towards the actors' acting skills is positive, and the overall evaluation is positive. Document-level and sentence pattern-level sentiment analysis cannot provide detailed information about the characteristics or attributes of the evaluation subject. To obtain such information, a more granular level is required.

## RELATED WORKS

The research on emotional orientation classification of film reviews has a long history. From the traditional document-level emotional classification research to the emotional classification of short texts of website reviews, emotional analysis has always been a very popular topic. In the text-based emotion classification algorithm, the researchers' work mainly focuses on the following aspects: semantic orientation-based method, traditional

machine learning-based emotion classification method, deep learning-based emotion classification method, *etc.*

The method based on semantic orientation calculates the positive and negative scores of all emotional words in the text to judge the document's emotional polarity, which is suitable for document-level and short-text emotional classification. The emotion dictionary plays an important role in emotion classification based on semantic orientation. *Kennedy & Inkpen (2010)* proposed an improved semantic orientation method to classify film reviews and considered the impact of context price fluctuation on emotion classification. *Quan & Ren (2010)* proposed a method combining emotion dictionary and syntactic dependency analysis for emotion classification and then relied on word-based similarity calculation to extract evaluation objects. *Gao, Xu & Wang (2015)* proposed a rule-based method to detect Chinese microblogs' emotional and emotional causes. First, they divided emotions into 22 types according to psychology. Then, according to the Chinese language's grammar rules, they customized a set of rules to detect the feelings and emotional causes of Chinese microblogs and calculated the score of each emotion and the corresponding emotional causes through conditional probability. The experimental results showed that the more grammar rules defined, the better the classification effect (*Gao, Xu & Wang, 2015*).

Most of the traditional emotion classification methods based on machine learning are based on supervised learning and usually require labeled corpora for classifier training. To achieve completely unsupervised classification, some researchers have improved the document topic generation model (LDA) to introduce emotion, which can detect emotion and topic from the text at the same time. It is unsupervised and does not require many labeled corpora. *Jo & Oh (2011)* proposed a new probabilistic model framework based on potential Dirichlet assignment (LDA) called the Aspect Emotion Unified model (ASUM). The author first proposed a theme model, Sentence LDA (SLDA), which assumes that all words in a single sentence are generated from one aspect, and then extended SLDA to the Aspect and Emotion Unified model (ASUM), which combines aspects and emotions to model emotions in different aspects. The results of this emotion classification show that ASUM is superior to other generation models and close to the supervised classification method (*Jo & Oh, 2011*).

Liang introduced a new technology called AS_LDA's emotion classification method assumes that words in subjective documents include two parts: emotion element words and auxiliary words, which are sampled according to emotion theme and auxiliary theme. Emotional element words include the goal of opinion, polar words, and modifiers of polar words. The experimental results showed that this method is superior to the potential Dirichlet distribution (LDA) (*Liang et al., 2014*). To associate potential themes with readers' induced emotions, *Rao et al. (2014)* proposed two emotional theme models, MSTM and SLTM. The evaluation of social emotion classification in the experiment verifies the effectiveness of the model presented in this paper. The social emotion dictionary samples generated further show that the model can find meaningful potential topics. Ghazi uses an SVM classifier to compare layered and common methods in emotion classification. Among them, neutrality, polarity, and emotion are divided hierarchically, and experiments showed that the layered method is superior to the ordinary method without considering

hierarchical information (*Ghazi, Inkpen & Szpakowicz, 2010*). To deal with the imbalance problem, *Wang et al. (2011)* proposed a multi-strategy set learning method to solve this problem. The integration method used multiple classification algorithms (naive Bayes, support vector machine and maximum entropy) to integrate the sample set, feature set and classifier set. *He (2013)* proposed three emotion classification methods based on machine learning (naive Bayes, SVM and SMO), used these methods to solve the problem of micro-blog emotion analysis problem, and compared each classification method's accuracy and performance. Moreo proposed document-level sentiment analysis in SVM and ANN. The experimental results showed that, except for some unbalanced data, the performance of ANN is better than that of SVM. Especially on the benchmark dataset of movie reviews, ANN outperforms SVM, even in the case of unbalanced data (*Moreo et al., 2012*).

Although the above research meets the requirements under specific conditions, it ignores the context of the text. It fails to fully exploit the sequence features of the text data, resulting in low accuracy of fine-grained emotion classification of film reviews. To carry out topic-based fine granularity emotion classification for film reviews, this study proposes a novel emotion classification model for film reviews based on the topic attention mechanism and dual channel LSTM model. Firstly, the Chinese movie review emotional lexicon and Word2vector tool are used to get a movie review emotional vector and semantic vector and input them into the LSTM network. Then, the special structure of LSTM is used to extract the sequence features contained in the movie review text sequence and complete the coding; Finally, the film theme attention mechanism is introduced for decoding, and the final emotion classification result is obtained through the softmax function of the full link layer and the output layer.

# TOPIC EXTRACTION AND VECTOR REPRESENTATION OF WORDS

## Extraction and representation of movie theme

A film review usually focuses on a particular film theme to express the feeling of watching the film. Usually, the words describing the film theme are nouns, such as special effects, plot, sound effects, etc. Each topic is an element to measure the actual meaning of the film. Its extraction in the corpus can be combined with a word clustering algorithm and the expertise of film experts. This research first uses a word clustering algorithm to cluster all nouns in the film review corpus, then uses the clustering center as the candidate subject words. Then, it combines the professional knowledge of film experts to make corrections so that the extracted subject words conform to the characteristics of the film field. The specific extraction process is shown in Fig. 1.

Afterword clustering and screening by film experts, the film theme set contains eight themes, namely, "plot," "special effects," "original work," "music," "thought," "theme," "acting," and "joke." The research paper projects each theme information of the movie into a multi-dimensional vector with continuous values. Each movie theme will have many related descriptive feature words or attribute words. This research will use the theme

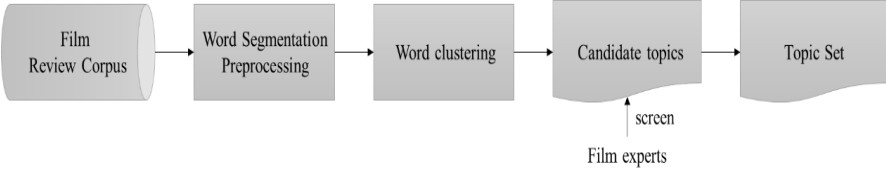

**Figure 1  File theme extraction process.**

**Table 1  Vector representation algorithm steps of movie theme.**

**Vector representation algorithm steps of movie theme:**

Input: The $i$th movie theme $T_i$
Output: the vector of the subject represents $\vec{V}_{t,i}$
1: The candidate feature word set consisting of $K$ nouns with high frequency in the film review corpus $HF \quad = \quad \{w_1, w_2 \dots w_k\}$
2: Calculate the $PMI$ values of $K$ candidate feature words and the $i$th movie theme $T_i$ respectively;
3: Set the threshold value to make the $PMI$ value greater than $\theta$ The candidate feature words of are screened out;
4: Build feature word set of film theme $T_i$ $F_i \quad = \quad \{w_1, w_2 \dots w_n\}$;
5: The average $PMI$ formula is used to calculate the film theme $T_i$ and the average $APMI_i$ of all feature word sets;
6: Vector representation of output topic $V_{t,i} = (0, 0, \dots, APMI_i, \dots, 0)$.

feature words to convert the movie theme into a vector representation form to facilitate the attention module to quickly lock the characteristics of a certain type of theme. $\vec{V}_{t,i} \in R^n$ represents the theme vector of the $i$th theme, and $n$ represents the vector dimension. Table 1 describes the vector representation algorithm of the movie theme.

## Vector representation of words

Using a word embedding algorithm to convert text information into word vectors is an important step in using deep learning to process text information. However, one of the major limitations of this algorithm is that words with similar structures but opposite emotional polarity are usually converted into similar semantic vectors, resulting in the loss of emotional information of words. To improve this limitation, this study describes a word from two aspects of semantics and emotion and inputs the emotion vector and semantic vector of the word into LSTM in a reasonable way to train a more efficient emotion classifier.

This research will use the Word2Vec tool to generate the semantic vector of words, take the movie review corpus as the training data, set the size of the co-occurrence window to 5, and set the dimension of the output vector to 150 dimensions, that is, each word will be converted into a 150-dimension semantic vector. Since most movie reviews are short texts, this study will take 100-word semantic vectors to represent a movie review. For movie reviews with less than 100 words, 150 dimensional all zero vectors will be used to supplement, to maintain the consistency of the length of the movie review sequence. So far, a movie review can be represented by 100 150-dimensional semantic vectors. A series

of semantic vectors of film reviews can be expressed in Formula (1).

$$TS_m = [\vec{v}_1, \vec{v}_2, .., \vec{v}_{100}]. \tag{1}$$

The semantic vector can only represent the semantic information of words. This study uses one lot-coding to construct a word emotional vector to improve the emotional information of words. It is found that the emotional tendency of a text includes not only positive and negative emotional words but also degree adverbs, negative words, interrogative words, etc. Therefore, this study will construct a 6-dimensional emotional vector of [positive, negative, neutral, interrogative words, and degree adverbs] according to the types of words that affect the emotional tendency of the text. The position of a certain type of word will be set to 1. The rest bits are set to 0. If "bad film" belongs to negative film review emotional words, its emotional vector can be expressed as [0, 1, 0, 0, 0, 0]. Suppose a word does not belong to any dictionary in the emotional lexicon of Chinese film reviews which belongs to an unlisted word. In that case, all bits of the emotional vector is set to zero, and the emotional vector sequence of a film review can be expressed in the form of the Formulas (2)–(3).

$$e_i^n = \begin{cases} 1, w \in \text{dictionary} \\ 0, w \notin \text{dictionary} \end{cases} \tag{2}$$

$$TS_e = [\vec{e}_1, \vec{e}_2, \ldots, \vec{e}_{100}]. \tag{3}$$

Here, $e_i^n$ represents the $n$th position of the emotional vector of the $i$th word, $TS_e$ represents the emotional vector of a film review. Like the semantic vector, the emotional vector of 100 words is also taken to represent the emotion of a film review. For film reviews with less than 100 words, a 6-dimensional zero vector will be used to supplement.

# EMOTIONAL ANALYSIS OF FILM REVIEWS BASED ON THEMATIC ATTENTION MECHANISM AND DUAL CHANNEL LSTM (DCLSTM-TAM)

## DCLSTM-TAM model construction

This research proposes a movie review emotion classification model (DCLSTM-TAM) based on topic attention mechanism and dual channel LSTM. The overall framework of this model is a coding and decoding structure. The coding part comprises the input and dual channel LSTM layers, and the decoding part comprises eight movie topic attention modules. First, based on the emotional lexicon of movie reviews constructed in this study, all words are coded into one pot, and the emotional vector of words is built. The semantic vector of words is obtained using the word2Vec tool as the input of the LSTM network. Then quantize the film theme information and construct multiple attention modules. Finally, the whole connection layer is used to classify the theme emotion of film reviews.

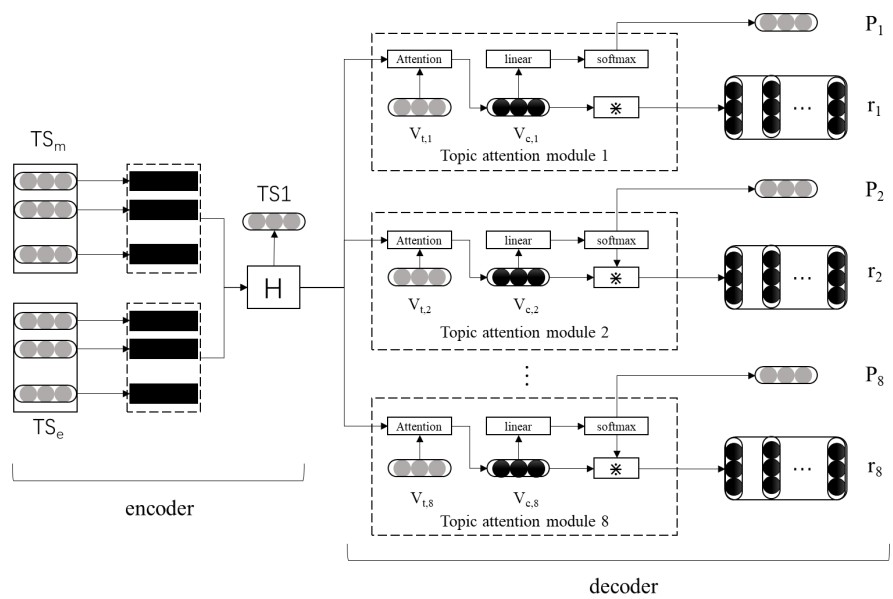

**Figure 2** **Network model diagram of DCLSTM-TAM.**

The specific thematic attention mechanism and the construction framework of the dual channel LSTM model are shown in Fig. 2.

(1) Output of encoder

It can be seen from Fig. 2 that the encoder mainly completes the extraction of semantic and emotional features of film reviews. First, the word2Vec tool and the emotion vector representation method proposed in this study complete the word embedding vector representation of film reviews. Then, the LSTM network is used for feature extraction to obtain the hidden layer state at each time point. For a film review with a word length of $w$, the hidden layer matrix output by the hidden layer can be expressed as $H = [\vec{h}_1, \vec{h}_2, \ldots, \vec{h}_t, \ldots, \vec{h}_m]$, where $\vec{h}_t$ represents the hidden layer state of the $t$ time step, and its dimension is the same as the output dimension of LSTM. Let $\vec{h}_t \in R^f$ have $f$ dimension, then $H \in R^{f*m}$. The encoder will input the hidden layer state matrix $H$ into each movie theme attention module respectively and generate the overall representation $TS_1$ of the movie review. Its calculation is shown in Formula (4).

$$TS_1 = \frac{1}{n} * \sum_{t=1}^{n} h_t \tag{4}$$

(2) Topic attention module calculation

It can be seen from Fig. 1 that the decoder mainly completes emotion classification based on film review topics and is composed of multiple topic attention modules. Each topic attention module includes attention, probability, and reconstruction. The number of theme attention modules here is the same as that of the extracted film review topics, which means that one film theme corresponds to one theme attention module. When the input film review involves multiple topics, the hidden layer state matrix H obtained by the

encoder will be sent to the corresponding topic attention module respectively. Different topic attention modules correspond to additional topic information, the film review topic vector. When the $\vec{v}_{t,i}$ The attention module model is trained to perform different attention calculations.

The function of the attention part is to enable the model to pay attention to the feature information of specific topics when extracting features from film reviews to obtain the text representation of specific topics of film reviews. First, the input hidden layer state matrix $H$ is spliced with a specific topic vector $\vec{v}_{t,i}$; Then, we calculate attention and obtain the text representation $\vec{v}_{c,i}$ with the $i$th film review theme through weighted average calculation. The specific calculation method of attention is shown in Formulas (5)–(6).

$$e_{t,i} = \tanh\left(W_{\alpha,i}\left[\vec{h}_t \cdot \vec{v}_{t,i}\right] + b_{\alpha,i}\right) \tag{5}$$

$$\vec{v}_{c,i} = \sum_{i=1}^{m} \frac{\exp\left(e_{t,i}\right)}{\sum_{k-1}^{m} \exp\left(e_{k,i}\right)} \vec{h}_t. \tag{6}$$

Here $W_{\alpha,i}$ is the weight matrix of the $i$th topic attention, and $W_{\alpha,i} \in R^{n+s}, b_{\alpha,i}$ is the offset term of the $i$th topic attention.

(3) Calculation of probability part

The probability part mainly calculates the probability that the text representation $\vec{v}_{c,i}$ of the $i$th film review topic belongs to a certain category. After the text representation $\vec{v}_{c,i}$ The review topic is input into the full connection layer of the film, and the probability of each category of the specific film review topic text to be classified can be calculated through the softmax function. This study has set three types of emotional classification: positive, negative, and neutral. The calculation method of the emotional classification probability of the related text of the $i$th film review topic is shown in Formula (7).

$$P_i = softmax\left(W_{q,i}\vec{v}_c, i + b_{q,i}\right). \tag{7}$$

Here, $P_i = \left[P_{i,1}, P_{i,2}, P_{i,3}\right]$ represents the probability that the emotional classification of the text related to the $i$th film review topic belongs to positive, negative, or neutral, $b_{q,\text{i}}$ and $W_{q,j}$ represents the weight and bias of the full connection layer, respectively.

(4) Calculation of reconstruction part

The primary function of the reconstruction part is to reconstruct the text representation $\vec{v}_{c,i}$ of the film review theme so that it can be a film review text representation including specific theme information, film review global information and emotional orientation information. The emotion classification probability generated by the probability calculation part is used to reconstruct $\vec{v}_{c,i}$. The reconstruction matrix of the specific topic $i$ generated is $R_i = \left[R_{i,1}, R_{i,2}, R_{i,3}\right]$, and its reconstruction calculation is shown in Formula (8).

$$R_{i,j} = p_{i,j}\vec{v}_{c,i}, j = 1, 2, 3. \tag{8}$$

### DCLSTM-TAM model training

This study uses the BP algorithm to train and optimize the DCLSTM-TAM model, and the minimum related loss function is constantly adjusted to optimize the model. The minimum loss function constructed in this study consists of two-fold loss functions $f_1(\alpha)$ and $f_2(\alpha)$. The function of $f_1(\alpha)$ is to maximize the emotional category probability of the actual topic text in the output of each topic attention module. The function of $f_2(\alpha)$ is to maximize the similarity between the reconstructed representation of the topic text of the actual emotion category and the overall representation TS of the film review in the output of each topic attention module. The specific calculation method is shown in Formula (9).

$$L(\alpha) = f_1(\alpha) + f_2(\alpha)$$

$$f_1(\alpha) = \sum_z \sum_i \sum_{j \neq c_{z,j}} \max\left(0, 1 - p_{i,c_{z,j}} + p_{i,j}\right) \tag{9}$$

$$f_2(\alpha) = \sum_z \sum_i \sum_{j \neq c_{z,j}} \max\left(0, 1 - TS_1 R_{i,c_{z,j}} + TS_1 R_{i,j}\right)$$

Where $z$ is the $z$th film review sample, and $c_{z,j}$ is the actual emotional category of the $z$th film review sample on the $i$th film theme.

The semantic vector of this study is initialized by word2Vec technology. The dimension of the semantic vector is set to 100 dimensions, the emotion vector is set to six dimensions, the output unit of the LSTM network is set to 200 dimensions, and the initialization of the movie theme vector and unlisted words uses a uniform distribution function to initialize randomly. A drop-out layer is added to the model to prevent overfitting during training, and its parameter is set to 0.5.

## RESULT ANALYSIS AND DISCUSSION

### Experiments and results on the emotional thesaurus of film reviews
*Experimental data set*

The experimental data set in this section mainly comes from the douban-crawler 22 containing the Chinese film review's emotional lexicon, including the film review test set and inspirational word test set. The film review test set includes 534,601 film reviews from 98 films, and 21,000 complete film reviews were screened after data preprocessing to remove Samsung film reviews and random comments with no apparent emotional tendency. The emotional classification and marking method of the experimental data set of film reviews use manual and machine marking methods. Film reviews with 1–2 stars in the film reviews are classified as negative emotional tendency reviews, and film reviews with 4–5 stars are classified as positive, inspirational tendency reviews. All the experimental data sets of film reviews are randomly divided into three groups, as shown in Table 2.

The emotional word test set includes the emotional word test in the film review field and the general emotional word test. The general emotional word test set includes 3,126 words, randomly divided into two test sets. The emotional word test set in the film review field

| Table 2 | Movie review experimental dataset. | | |
|---|---|---|---|
| **Name** | **Positive emotion** | **Negative emotion** | **Total** |
| Film Review Test Set A ($F_A$) | 3,205 | 2,795 | 6,000 |
| Film Review Test Set B ($F_B$) | 4,000 | 4,012 | 8,012 |
| Film Review Test Set C ($F_C$) | 3,586 | 3,402 | 6,988 |

| Table 3 | Emotional word test set. | | |
|---|---|---|---|
| **Name** | **Positive emotion** | **Negative emotion** | **Total** |
| General Affective Words Test Set a ($W_a$) | 886 | 914 | 1,800 |
| General Affective Words Test Set b ($W_b$) | 626 | 700 | 1,326 |
| Test set of emotional words in film review field ($W_c$) | 54 | 46 | 100 |

is obtained by manually counting and classifying positive and negative emotions from the film review corpus, including 100 emotional words. See Table 3 for the specific emotional word test set.

### The influence of emotional thesaurus in the film review field on the emotional classification of film reviews

To verify the effect of the emotional lexicon in the field of film reviews built in this study on improving the emotional classification of film reviews, we will use three emotional dictionary methods to classify emotions based on the experimental data set of film reviews. Methods 1. The Chinese emotional vocabulary ontology was used as the reference dictionary for computing the emotions named M1. The Chinese emotional lexicon ontology and the emotional lexicon in film review were used as references for emotional calculation named M2. The emotional lexicon in film review was used as a reference for emotional calculation named M3. Then the emotional polarity value of emotional words in a film review is counted and accumulated. If the value is less than 0, the film review is negative; otherwise, it is positive. The experimental results of film review emotion classification based on different emotion dictionaries are shown in Fig. 3.

The above experimental results show that the emotional lexicon of film reviews constructed in this study can significantly improve the accuracy of the emotional classification. Among them, M3 classifies emotions based on the lexicon. The accuracy with M3 is 84.93% on the three test sets, which is significantly better than the M1 and M2. The emotional classification of degree and negative word dictionaries can also improve the classification accuracy to a certain extent, which shows that the emotional lexicon constructed in this study has practical and effective value.

### Related experiments and results of the DCLSTM-TAM model
#### Experimental data set

The experimental data set in this section mainly comes from 6,800 watercress film reviews crawled by douban-crawler 22, divided into training data sets (3,900) and test data sets

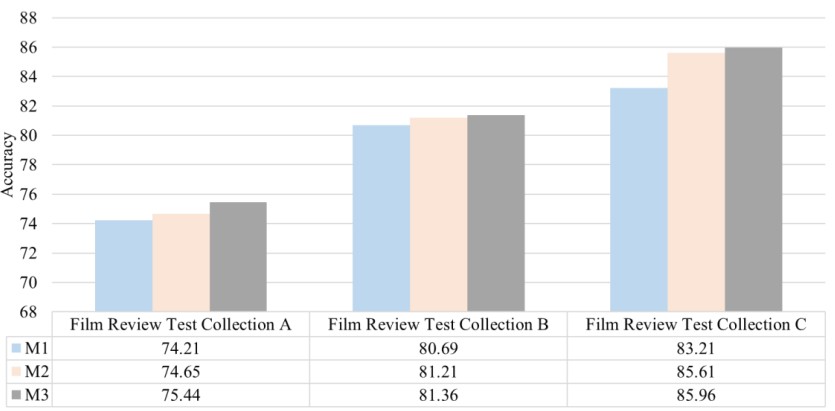

**Figure 3** Classification accuracy based on different sentiment dictionaries.

(2,900). The emotional classification and marking method of the film review experimental data set adopts a combination of manual and machine marking. Film reviews with a score of 1–2 stars are classified as film reviews with negative emotional tendencies, film reviews with a score of 3 stars are classified as neutral film reviews, and film reviews with a score of 4–5 stars are classified as the film reviews with positive emotional tendencies. Then, two groups of markers are asked to manually classify the emotional tendencies of each theme of each film review, and the classification results are counted many times until the error of the two groups of marking personnel is 0. Each film review will involve one or more film themes. See Table 4 for the statistics of specific experimental data sets.

### The influence of the DCLSTM-TAM model on the emotional classification of film reviews

To verify the rationality of the emotion vector and semantic vector input into LSTM through dual channel fusion proposed in this study and the impact of adding emotion vector on the accuracy of film review emotion classification. This research trains different LSTM networks based on the training set and test set of film reviews and classifies the emotions of film reviews. The specific comparative experimental methods are as follows:

Method 1 (LSTM): The traditional LSTM method is only a semantic vector + LSTM.

Method 2 (DCLSTM): semantic vector + emotional vector + LSTM, and the fusion method of the two vectors is to input LSTM respectively in a dual channel way, and the fusion is realized before the full connection layer is input.

Method 3 (DCLSTM-TAM): Film review emotion classification model based on topic attention mechanism and dual channel LSTM.

In this study, the accuracy rate A (accuracy) of emotion classification is taken as the experimental indicator. The accuracy changes in emotion classification of film reviews based on different LSTM networks are shown in Fig. 4.

This study compares the experimental results of the constructed DCLSTM-TAM model with the classical emotion classification model LSTM and the DCLSTM model with emotion vector. It comprehensively evaluates the model's performance using three

**Table 4  Movie review & theme sentiment classification experimental data set.**

| Film theme | Positivity | | Neutral | | Negativity | |
|---|---|---|---|---|---|---|
| | Training set | Test set | Training set | Test set | Training set | Test set |
| Plot | 1,021 | 988 | 512 | 302 | 1121 | 865 |
| Special effects | 311 | 272 | 421 | 212 | 512 | 412 |
| Original | 221 | 252 | 102 | 81 | 212 | 103 |
| Music | 541 | 412 | 421 | 312 | 312 | 210 |
| Thought | 321 | 291 | 103 | 75 | 59 | 26 |
| Theme | 198 | 161 | 98 | 75 | 161 | 102 |
| Acting skill | 827 | 412 | 798 | 521 | 592 | 411 |
| Laughing stock | 211 | 103 | 421 | 216 | 312 | 293 |
| Total | 3,651 | 2,891 | 2,876 | 1,794 | 3,281 | 2,422 |

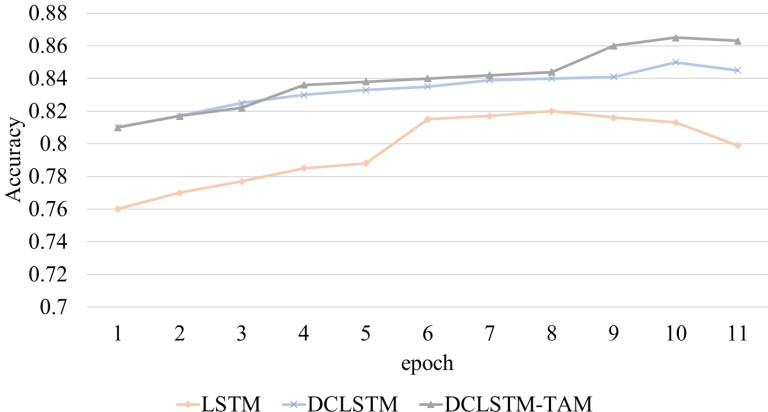

**Figure 4  The accuracy-epochs curve of different classification models.**

indicators: accuracy (P), recall (R) and F1 value. Among them, the accuracy rate P (precision) represents the proportion of correctly predicted positive samples in the actual predicted positive samples, the recall rate R (recall rate) represents the proportion of correctly predicted positive samples in the actual positive samples, and the F1 value is the evaluation index of the comprehensive accuracy rate and recall rate. This study uses these three indicators to evaluate the model's performance, and the relevant calculation formula is shown in Formula (10).

$$P = \frac{TP}{TP + FP}$$

$$R = \frac{TP}{TP + FN}$$

$$A = \frac{TP + TN}{TP + TN + FP + FN} \times 100\% \tag{10}$$

Table 5 Comparison of F1 using different classification methods.

| Model | Experimental indicators | Classification | | |
|---|---|---|---|---|
| | | Positivity | Negativity | Neutral |
| | P | 0.837 | 0.812 | 0.542 |
| LSTM | R | 0.881 | 0.831 | 0.19 |
| | $F_1$ | 0.858 | 0.821 | 0.530 |
| | P | 0.892 | 0.869 | 0.581 |
| DCLSTM | R | 0.925 | 0.835 | 0.561 |
| | $F_1$ | 0.908 | 0.852 | 0.571 |
| | P | 0.924 | 0.884 | 0.683 |
| DCLSTM-TAM | R | 0.931 | 0.908 | 0.701 |
| | $F_1$ | 0.927 | 0.896 | 0.692 |

$$F_1 = \frac{2*P*R}{P+R}.$$

The specific experimental results are shown in Table 5.

$F_1$ with different classification methods of the comparison results, as shown in Fig. 5. It can be seen from the results of three classification experiments that the DCLSTM model with emotion vector improves the A value of positive, negative and neutral emotion classification by 5.0%, 3.1% and 4.1%, respectively, compared with the traditional LSTM network with only semantic vector input, which indicates that the LSTM network with emotion vector can better capture emotion information; The DCLSTM-TAM model proposed in this study is applied to the classification of positive, negative and neutral emotions. In terms of 1 value, it is 6.9%, 7.5% and 16.2% higher than the traditional LSTM network which only inputs semantic vectors, respectively. It not only improves the classification efficiency of positive and negative emotional tendencies but also significantly improves the classification efficiency of film reviews with less emotional tendencies. It shows that the use of multiple film theme attention modules can fully extract the hidden features of specific film review topics so that the model has a better effect on the emotional classification of film review texts, which verifies the effectiveness of the model built in this study to a certain extent.

This study tested the effectiveness of the proposed algorithm in building an emotional dictionary. The role of the constructed emotional dictionary in a film review, in the emotional classification of film review texts, and the effect of topic attention mechanism and dual channel LSTM model.

# CONCLUSIONS

This research adopts deep learning networks and other methods to achieve the automatic emotional classification of film reviews, significantly improving the accuracy of emotional classification of film reviews. It also reduces the cost of artificial emotion classification and provides help for the investment decision-making of cinemas and the choice of customers to watch films. The Chinese semantics are complex, the available emotional dictionary

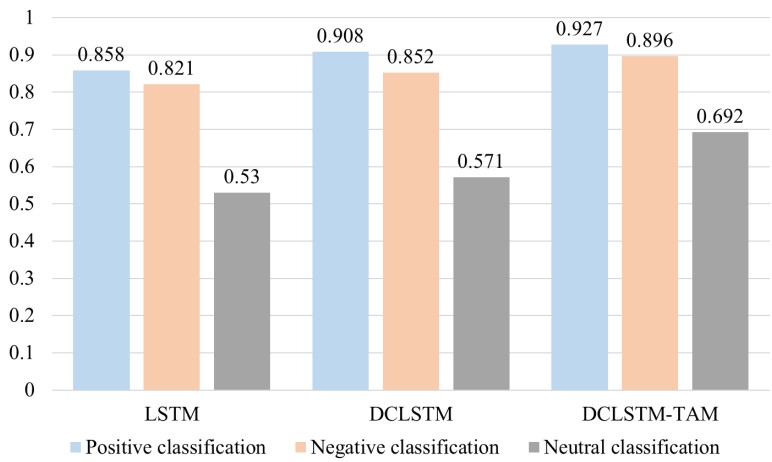

**Figure 5   Comparison of F1 using different classification methods.**

resources in the film review field are few, and the domestic emotional classification started late. The domain characteristics of the film review emotion classification make the traditional model unsuitable. Therefore, this paper constructs an emotional lexicon and constructs an improved model based on topic attention mechanism and dual channel LSTM; the main contributions of this study include:

(1) In this study, a word clustering algorithm is used to cluster all nouns in the film review corpus, and the clustering center is used as the candidate subject words and then combined with the professional knowledge of film experts to modify so that the extracted subject words conform to the characteristics of the film field;

(2) The DCLSTM-TAM based on topic attention mechanism and dual channel LSTM is proposed in this study. The semantic vector constructed for each word based on the emotional lexicon in the field of film review makes the input layer of the LSTM model input not only the semantic vector but also the emotional vector and constructs a topic attention mechanism for decoding so that DCLSTM TAM model can fully extract the hidden features of specific film review topics to classify the emotions of each topic of a film review, The experimental results show that the DCLSTM TAM model improves the classification efficiency of both positive and negative emotional tendencies and the classification efficiency of film reviews with less emotional tendencies.

Although the film review emotion analysis method based on deep learning proposed in this study has achieved some results, due to the constraints of time, technical scope and other aspects, the research in this direction can be further improved as follows.

(1) Because the data set annotation depends on the scoring data, there are some inaccuracies. In the subsequent research, a part of manual correction can be added.

(2) It only classifies the overall emotional orientation based on the review data, but fails to analyze the emotional classification of each sentence in terms of theater, plot, special effects, *etc.* Later, it can be further discussed according to the aspect-based emotional

analysis research to find out which specific aspects or features of the film the audience is more interested in.

In the future, the proposed model can also provide a theoretical basis for other relevant text emotion classification (such as microblog topics, commodity reviews, and hotel reviews) to facilitate more in-depth research on text emotion classification in different fields.

## ACKNOWLEDGEMENTS

The author of this article thanks the anonymous reviewers whose comments and suggestions helped improve this manuscript.

### Funding
The author received no funding for this work.

### Competing Interests
The author declares that they have no competing interests.

### Author Contributions
- Yufei Wang conceived and designed the experiments, performed the experiments, analyzed the data, performed the computation work, prepared figures and/or tables, authored or reviewed drafts of the article, and approved the final draft.

### Data Availability
The code is available in the Supplemental File. The data is available from GroupLens: https://grouplens.org/datasets/movielens/.

### Supplemental Information
Supplemental information for this article can be found online at http://dx.doi.org/10.7717/peerj-cs.1295#supplemental-information.

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
