# Peer review of "Research on emotion classification technology of movie reviews based on topic attention mechanism and dual channel long short term memory"

_PeerJ Computer Science, doi:10.7717/peerj-cs.1295_

## Round 0.1 · original submission · Minor Revisions

Dear Author,

Thank you for your submission to our esteemed journal. Your paper has been reviewed by the experts in the field and myself. We feel that the paper has merits but it needs couple of change before we consider it further. Therefore, please carefully revise the paper according to reviewers comments and re-submit. Thank you

Reviewer 1 ·

Basic reporting

The paper is generally well-designed and well-written. However, it is suggested to add a few recent references to justify the problems and motivation.

Experimental design

It is also suggested to compare the results of the LSTM algorithms with any counterpart to justify the contribution.

Validity of the findings

The simulation results should be compared with the recent findings in the literature review.

Additional comments

The emotional classification of film reviews has high research value. However, there are few emotion dictionaries and analysis tools available for reference and use in the field of film criticism, and the accuracy of emotion classification needs to be improved. This research introduces attention mechanism and dual channel LSTM while building an emotional dictionary in the field of Chinese film reviews, classifies Chinese film reviews into fine-grained emotional categories in terms of topics, effectively handles film reviews with vague emotional tendencies, improves the accuracy of emotional classification of film reviews, and has certain innovation. However, there are still some problems to be improved:

A. In the Related works section, the author introduces the related research of text-based emotion classification algorithms and asks the author to survey the main aspects of the researchers' work.

B. In the Related works section, the author introduced many related types of research on text-based emotion classification algorithms, but the author did not explain the main research ideas of this study. Please supplement this section.

C. In the process of extracting and representing the movie theme, the author has extracted the theme words that meet the characteristics of the movie field. Please introduce the specific extraction flow chart.

D. In the process of extracting and representing the movie theme, the author uses topic feature words to convert the movie theme into a vector representation. Please introduce the steps of this algorithm.

E. During the training of the DCLSTM-TAM model, what are the dimensions of the semantic vector, emotional vector and network output unit set by the author?

F. In the Conclusions section, the author summarizes the key points and contributions of the study. Is there any deficiency in the study?

Reviewer 2 ·

Basic reporting

(1) In the introduction part, the author proposes to conduct emotional analysis on unstructured texts such as user evaluations and comments. According to the technical characteristics, it can be divided into three categories: rule-based emotional thesaurus matching analysis method, statistical machine learning based emotional classification method, and machine learning method classification combined with dictionary filtering emotional analysis algorithm, and introduces the related research results, What are the shortcomings of these studies in practical applications?

(2) In the Related works section, the author introduces the related research of text-based emotion classification algorithm. Are there any deficiencies in these research that need to be improved?

Experimental design

(1) In the process of vector representation of words, why does the author describe a word from both semantic and emotional aspects? What are the limitations of using word embedding algorithm?

Validity of the findings

(1) When explaining the impact of DCLSTM-TAM model on the emotional classification of film reviews, the author uses accuracy (P), recall (R) and F_ The performance of the model is comprehensively evaluated by the three indicators of 1 value. The author is requested to evaluate the accuracy (P), recall (R) and F1 Explain the definition and calculation method of the three indicators.

(2) In the Inclusions section, the author summarizes the innovation points of the study. What is the next research direction of the study?

Additional comments

Aiming at the massive film review data, this research adopts deep learning network and other methods to achieve automatic emotional classification of film reviews, greatly improving the accuracy of emotional classification of film reviews, reducing the cost of artificial emotional classification, and providing help for the investment decision-making of cinemas and the choice of customers to watch films. Moreover, the emotion classification method and model proposed in this study have good transferability, and the change of training corpus is also applicable to other short text fields. Although this paper has some innovation, there are still some details to be explained or improved.

---

## Round 0.2 · accepted · Accept

Dear author,
Thank you for your fine contribution to our esteemed journal. we hope you have enjoyed your experience while publishing with us.

Reviewer 1 ·

Basic reporting

The paper is well-written and organized this time.

Experimental design

The experimental design, setup, details, and implementation are well explained in this version.

Validity of the findings

The key findings are well justified in relation to the model and methodology.

Additional comments

In my opinion, the paper can be accepted in its current form.

Reviewer 2 ·

Basic reporting

All changes have been completed.

Experimental design

All changes have been completed.

Validity of the findings

All changes have been completed.

Additional comments

All changes have been completed.